



**Do marine benthos breathe what they eat?**
Xiaoguang Ouyang, Cheuk Yan Lee, Shing Yip Lee
Simon F.S. Li Marine Science Laboratory, School of Life Sciences, The Chinese University of
Hong Kong, Shatin, Hong Kong SAR, China
Correspondence to:
Xiaoguang Ouyang (email: x.ouyang@cuhk.edu.hk)
Shing Yip Lee (email: joesylee@cuhk.edu.hk)



**Abstract**
Intertidal benthos link tertiary predators and primary producers in marine food webs as well as
directly contribute to sediment $CO_2$ emission. However, current methods for studying food
sources of marine benthos are time-consuming and does not allow direct estimates on feeding
regime-related $CO_2$ production. We examined the foods of mangrove crabs and gastropods as
well as their corresponding $CO_2$ production by using cavity-ring down spectroscopy to measure
the $\delta^{13}C$ of consumer-respired $CO_2$, considering the effects of feeding regime, benthos taxa, and
dominant feeding habit. Benthos taxa and feeding habit have significant impact on $\delta^{13}C$ of
respired $CO_2$. Particularly, the $\delta^{13}C$ of crab (-23.9±0.4‰) respired $CO_2$ was significantly lower
than that from gastropod (-17.5±1.3‰) respiration. The $\delta^{13}C$ of deposit-feeder respired $CO_2$ was
significantly higher than that from detritivores. There are significant differences in the amount of
$CO_2$ emitted and $\delta^{13}C$ of crab respired $CO_2$ under different feeding regimes. The differences
reflect diet-switching and fuel-switching in the crabs, i.e. 'you breathe what you eat'. Significant
differences in $CO_2$ production of crabs also exist between those feeding on microphytobenthos
(0.13±0.02 mmol $g^{-1}$ $day^{-1}$) and on field collection (0.31±0.03 mmol $g^{-1}$ $day^{-1}$). $CO_2$ production
of crabs is strongly related to carapace width and length. The $\delta^{13}C$ of respired $CO_2$ from
mangrove crabs reflects their diet while crab-respired $CO_2$ flux is related to crab size. These
relationships enable partitioning the feeding habit and food sources of key benthos, and help
incorporate their contribution into the overall sediment-atmosphere $CO_2$ fluxes in mangroves.



## 1 Introduction

Intertidal benthos are well known to play different roles in processing nutrients in the intertidal food webs depending on their taxonomic affiliation and feeding habit. In mangroves, sesarmid crabs are reported to use mangrove leaf litter as the most important carbon source, with diatoms or live/dead animal prey as the dominant nitrogen source, while ocypodid crabs and gastropods are mainly deposit feeders, preferring the microphytobenthos (Kristensen et al., 2017; Lee, 2008). The different feeding habits (e.g. detritivorous or deposit-feeding) of the mangrove benthos may result in differences in their metabolic processes. Past mangrove studies revealing the food sources of the mangrove benthos have advanced from earlier gut analysis to stable isotope analyses (Lee et al., 2014). Stable isotope analysis unravels benthic food sources by extracting benthos muscle tissues and analyzing tissue and food stable isotope values of carbon, nitrogen and occasionally sulfur (Bui & Lee, 2014; Chong et al., 2001). However, dietary analysis is a time-consuming process that often requires close monitoring of individual predators or collection and analyses of prey remains in the gut or faeces of the predator (Caro, 1994; Wachter et al., 2012).

In contrast to traditional stable isotope analyses of predator tissues, $\delta^{13}C$ of predator-respired $CO_2$ have increasingly been applied to study the food sources and feeding habit of predators. The latter approach has advantages over isotope analyses of predator tissues because it can provide information about both the most recently consumed diet and the integrated diet over longer periods (Engel et al., 2009). While some adults of aerial and aquatic migratory species occupy intertidal habitats during specific seasons of the year, others are permanent inhabitants (Vernberg, 1993). The variable feeding habits of marine migratory species make it useful to study both their immediate diet and integrated diets. In contrast, some tissues (e.g. muscles) have





slower turnover rates and their isotopes can only reflect the integrated diet over longer periods
(Carleton et al., 2004). Further, breath $\delta^{13}C$ can be repeatedly measured non-destructively for the
same predator and thus can track the changes in its food sources. Carleton et al. (2004) explored
the use of $\delta^{13}C$ of hummingbirds exhaled $CO_2$ to demonstrate their shift from a C3 to C4 diet.
Voigt et al. (2008) found free-ranging vampire bats prefer cattle blood by analyzing $\delta^{13}C$ of their
respired $CO_2$. Its application was also tested in other animals such as grasshoppers (Engel et al.,
2009). $\delta^{13}C$ of predator-respired $CO_2$ is sufficient for quickly unravelling the prey consumed
without the involvement of diet and predator tissue $\delta^{13}C$ values. However, this approach still
remains uncommon in marine studies for examining the food of marine benthos.

Moreover, the measurement of benthos-respired $CO_2$ and their isotope signatures does not

only reveals their food sources but also helps assess their contribution to sediment respiration
after adjusting for the effects of feeding habit and regime (e.g. active vs. dormant status, and
feeding on microphytobenthos vs. yellow leaves). $CO_2$ emission rates from the sediment surface
of mangroves are more than doubled if the contribution of marine benthos and their burrows is
included (Kristensen et al., 2008; Ouyang et al., 2017; Penha-Lopes et al., 2010). Some crabs
may remain dormant most of the time, with a short active period (e.g. 90 days, Katz (1980))
during the year. Should the starved condition during dormant periods be unaccounted for, it may
lead to erroneous estimations on the animals' contribution to $CO_2$ emission from the sediment
surface. Further, it is well established that C3 and C4 plants have distinct isotope signatures (Fry,
2006), which may result in differences in $\delta^{13}C$ of benthos-respired $CO_2$. Nonetheless, few studies
explore whether the increase in $CO_2$ emission rates due to marine benthos is related to feeding
regime or their feeding habit.



Cavity ring-down spectroscopy (CRDS) is one of the most recent advances in measuring the
concentration as well as isotopic values of many biogenic gases, including $CO_2$, $CH_4$, and other
gases that have unique near-infrared absorption spectra. CRDS works by quantifying the effect
of this absorption (Welch Jr et al., 2016). It has been used in marine studies to partition sources
of ecosystem and sediment respiration of mangrove seedlings (Ouyang et al., 2018), and the
variation of $\delta^{13}C$ of greenhouse gases emitted from estuaries and other ecosystems (Jacotot et al.,
2018; Munksgaard et al., 2014; Rosentreter et al., 2018; Sea et al. 2018; Ouyang et al., 2020).
We conducted the first study using CRDS to relate the feeding regimes of different intertidal
benthos with $\delta^{13}C$ of their respired $CO_2$. We also measured the different isotope enrichment
between marine benthos-respired $CO_2$ and their diets for different benthos taxa and feeding
habits. The relationship between benthos size and feeding regime to their $CO_2$ production was
also assessed. Intertidal benthos are hypothesised to "breathe what they eat" in terms of both
breath $\delta^{13}C$ and $CO_2$ production. We explored the influence of benthic taxa, feeding regime and
feeding habit on the $\delta^{13}C$ of benthos-respired $CO_2$ and/or benthos $CO_2$ production (i.e. $CO_2$
respired per unit mass per day), and the relationship between benthos $CO_2$ production and their
body sizes through laboratory experiments. We put forward the novel paradigm 'You Breathe
What You Eat', which adds to the well-known paradigm of 'You Are What You Eat' in food
web studies. This study will inform future efforts to estimate the contribution of different food
sources to the diet of mangrove benthos over both short and long periods.

**2 Materials and Methods**
**2.1 Sample collection**





We collected small (2-4 cm carapace width) but numerically dominant brachyuran crabs
(including sesarmids, varunids and ocypodids) and gastropods from the mangrove forests in Mai
Po Nature Reserve (22°30'N,114°02'E) and Ting Kok (22°28'N,114°13'E), Hong Kong.
Senescent leaves of the mangrove *Kandelia obovata* were hand-picked from the trees. The leaves
can be identified by their yellow colour and easily detachable from the branches. Surface
sediments (down to 1cm) were collected by a syringe with the needle end removed. The samples
were kept on ice before transportation to the laboratory.

**2.2 Sample pre-treatment and separation**
Each animal was kept in a small container with seawater covering the bottom to avoid
desiccation. The animal food items were treated and/or purified before the consumption
experiments. Upon return to the laboratory, the leaves were immersed in seawater for around 24
hours to allow leaching of deterrent chemicals such as tannins, which may deter crabs from
feeding. Microphytobenthos (MPB), mainly in the form of diatoms, were separated from the
surface sediments by the 'sieve and spin' method (Bui & Lee 2014). Specifically, the sediments
were suspended in seawater and sieved through a 45μm mesh. The filtrate was further passed
through a 5 μm filter, and the residue was resuspended in Ludox colloidal silica (Sigma, density
1.34 g ml$^{-1}$). Then the mixture was vortexed and centrifuged at 4000 rpm for 10 minutes. The
microphytobenthos concentrated in a distinct layer at the top of the colloidal silica were then
separated via a pipette, before confirming to be predominantly microphytobenthos using a
microscope. The MPB was washed in Milli Q water to remove the remnant colloidal silica and
washed again when the microphytobenthos had settled down. This process was repeated several



times until the water was clear. The MPB were then collected on pre-combusted GF/F filters
(Whatman).

**2.3 Experimental design**

We conducted a series of experiments to examine factors influencing C isotopic signatures

and/or $CO_2$ production of the mangrove benthos: (1) the effect of benthic taxa (i.e. crabs vs.
gastropods); (2) the effect of feeding regime (on collection vs. fasted, feeding on yellow
mangrove leaves vs. diatoms). Mangrove benthos "on collection" and "fasted" correspond to the
active and dormant status of benthos, respectively; (3) the effect of feeding habit (i.e. deposit
feeders vs. detritivores); and (4) the relationship between $CO_2$ production and animal body size.
Thirty animals were used each in experiments (1) and (3), while 10 sesarmid crabs were used in
experiment (2), and 25 animals (sesarmid crabs and gastropods) in experiment 4.
In experiment (1), the animals were put in plastic 0.8 l containers covered by aluminium foil to
minimise disturbance. There is a small hole on the lid of each container to keep the pressure
balance between the inside and outside of the containers. A soft plastic hose was connected to
the lid of each container with the other end closed by a stop-cock. Twenty minutes after closing
each container, a syringe was connected to the other outlet of the stop-cock, and 30 ml of gas
was collected from the container. The needle end of the syringe was also closed by a stop-cock
until analysis. Gases were collected over a period of 50 minutes (five times, every 10 minutes).
In experiment (2), gas samples were collected similarly collected upon arrival at the laboratory
as well as under starving conditions each day for three days to simulate the dormant status.
Afterwards, they were fed with mangrove leaves and then, after fasting for 3 days, fed with
MPB. Gas samples were collected separately when the crabs were fed on different foods. In





experiment (3), the benthos were fed treated yellow leaves or MPB, depending on their main
feeding habits. Deposit feeders, including gastropods (e.g. *Terebralia sulcata*), ocypodid crabs
(e.g. *Uca arcuata*) and varunid crabs (e.g. *Metaplax longipes*), were fed MPB. Detritivores,
including sesarmid crabs, e.g. *Parasesarma bidens* and *P. pictum*, were fed yellow mangrove
leaves.

**2.4 Sample analysis and measurement**
In experiment (4), the carapace width (CW) and length (CL) of crabs and shell length of
gastropods were measure by Vernier callipers. After measuring $CO_2$ production as described
above, the crabs were sacrificed and dried at 60°C until constant weight. Subsamples of dried
leaf and MPB samples (around 5 mg) were weighed into tin capsules for stable isotope analysis.
Another group of crabs were dissected and muscle tissues extracted, dried and prepared in the
same way for stable isotope analysis. Their elemental contents (carbon and nitrogen), $\delta^{13}C$ and
$\delta^{15}N$ values were analysed by a EuroVector Elemental Analyser - Nu Perspective Isotope Ratio
Mass Spectrometer (IRMS) at The University Hong Kong, with iACET standards used for
quality control check.

$CO_2$ and $CH_4$ concentrations and $\delta^{13}C$ values of the gas samples were measured by a Picarro

G 2201-i CRDS analyser (Picarro Inc., USA). The syringes were connected to the inlet of CRDS
analyser to allow gas to be sucked into the analyser by a vacuum pump. The analyser measured
$\delta^{13}CO_2$, $CO_2$ and $CH_4$ concentrations at an interval of five seconds with guaranteed precision of
<0.012‰, 200ppb + 0.005% and 50ppb + 0.05%, respectively. Standard gases of mixed $CO_2$
(1008 ppm) and $CH_4$ (10.2 ppm) (ARK NIC, West Indies) were used to check the accuracy of
the analyser.



Keeling plots (Keeling, 1961) were used to estimate $\delta^{13}C$ of crab-respired $CO_2$. Keeling plots
assume that $\delta^{13}C$ of $CO_2$ within a space reflects the mixture of some amount of $CO_2$ contributed
by sources in the system and some background amount of $CO_2$ that is already present in the same
space (Dawson et al., 2002). It is described by the isotope mixing model and mass balance model
as below:

$$\delta^{13}C_{system}[CO_2]_{system}=\delta^{13}C_{atm}[CO_2]_{sample} + \delta^{13}C_{source} [CO_2]_{source} \qquad (1)$$


$$[CO_2]_{system}=[CO_2]_{sample}+[CO_2]_{source} \qquad (2)$$

where $\delta^{13}C_{system}$, $\delta^{13}C_{source}$, $\delta^{13}C_{atm}$ are the $\delta^{13}C$ values for the system, the source and sample
space, respectively. $[CO_2]_{system}$, $[CO_2]_{source}$ and $[CO_2]_{sample}$ are the concentrations for the system,
the source and sample space, respectively. The equations (1) and (2) can be combined and
rearranged as below

$$\delta^{13}C_{system}=\delta^{13}C_{source}+[CO_2]_{sample}(\delta^{13}C_{sample} -\delta^{13}C_{source} )\frac{1}{[CO_2]_{system}} \qquad (3)$$

The same principle can be applied to measure $\delta^{13}C$ of benthos- and other animal-respired $CO_2$,
as discussed elsewhere (Carleton et al., 2004). As indicated in equation (3), the y-intercept of a
linear regression of $\delta^{13}C_{system}$ against $\frac{1}{[CO_2]_{system}}$ provides an estimate $\delta^{13}C$ of benthos-respired
$CO_2$. $[CO_2]_{source}$ was estimated as the slope of the regression relationship between $[CO_2]_{system}$
(t=0, 10, 20, 30 and 40 minutes) and t. A conceptual model was used to describe the process of
measuring benthos $CO_2$ production and $\delta^{13}C$ of respired $CO_2$ by CRDS (Fig. 1)
The containers used to collect greenhouse gases respired by the animals are similar to a static
chamber, for which the respired greenhouse gas flux (mmol day$^{-1}$) was standardized across the
diet treatments and estimated as below



$$F = V \left(\frac{d[CO2]source}{dt}\right) \frac{1}{V_0} \frac{P}{P_0} \frac{T_0}{T} \qquad (4)$$
Where V is the chamber volume subtracting the volume of the crab. $\frac{d[CO2]source}{dt}$ is usually taken
to be the slope of the linear regression of $[CO_2]_{source}$ on t (Rolston, 1986). Here t is the series of
gas collection time from the start to end, i.e. every 10 minutes for five times. $V_0$ is the gas molar
volume under standard conditions. $P_0$ and $T_0$ is the standard pressure and temperature. P and T is
the pressure and temperature in the container.

**2.5 Statistical analysis**
We conducted a two-way analysis of variance (ANOVA) to examine the impact of benthos
taxa and feeding habit on $\delta^{13}C$ of benthos-respired $CO_2$. When significant treatment effects were
found, Tukey's HSD test was used to detect difference among groups. We also conducted a one-
way ANOVA to examine the impact of feeding regime on $\delta^{13}C$ of benthos-respired $CO_2$ and
benthic $CO_2$ production. Before ANOVA, the assumptions of normality and homoscedasticity
were tested ($\alpha=0.05$). Normality was tested using the Shapiro-Wilk normality test.
Homoscedasticity was tested using the Levene test. Linear regression was used to examine the
relationships between crab $CO_2$ production and carapace width/length/ weight, as well as
between gastropod $CO_2$ production and shell length/weight. The assumption of normality was
tested as described above. Given only significant relationships were found for sesarmid crabs,
student's t test was performed to test the difference in the ratios of carapace length and width
between ocypodid crabs and sesarmid crabs, to show if the relationship between $CO_2$ production
of sesarmid crabs and crab size applies to ocypodid crabs. Student's t test was also used to
compare (1) the difference in C/N ratios; (2) $\delta^{13}C$ and $\delta^{15}N$ values between mangrove yellow
leaves and the microphytobenthos; and (3) $\delta^{13}C$ of $CO_2$ respired by crabs on collection and that



of muscle tissues. Data are presented as mean±standard error (SE). All the data analyses were
conducted using R programming language (R Core Team, 2014). The R package 'car' was used
to conduct ANOVA (Fox & Weisberg, 2011).

**3 Results**
**3.1 The sources of variations of $\delta^{13}C$ of intertidal benthos respired $CO_2$**
Our results show that $\delta^{13}C$ of intertidal benthos respired $CO_2$ varied significantly with benthos
taxa, dominant feeding habit and/or feeding regime. Benthos taxa (1) and dominant feeding habit
(2) had a significant influence on $\delta^{13}C$ of benthos-respired $CO_2$ (ANOVA, $F_1$=48.4, P<<0.001;
$F_2$=12.9, P<<0.001). Further post-hoc analyses showed that the $\delta^{13}C$ of crab (-23.9±0.4‰)
respired $CO_2$ was significantly lower than that of gastropod (-17.5±1.3‰) respired $CO_2$ (Tukey's
HSD test, P<<0.001, Fig. 2a). The $\delta^{13}C$ of deposit-feeders (e.g. *Terebralia sulcata* and *Uca*
*arcuata*) respired $CO_2$ (-19.8±0.8‰) was significantly higher than those of detritivores (e.g.
*Parasesarma bidens*) respired $CO_2$ (-24.7±0.3‰, Tukey's HSD test, P=0.004, Fig. 2b).
There were also significant differences in $\delta^{13}C$ of crab-respired $CO_2$ among different feeding
regimes (ANOVA, F=5.4, P<0.001, Fig. 3). In particular, $\delta^{13}C$ of crab-respired $CO_2$ was
significantly lower when crabs were fed on leaves (-26.6±0.3‰) than on MPB (-23.8±0.4‰,
P<0.001), and on collection (-24.8±0.6‰, P<0.05). This is consistent with the diet $\delta^{13}C$ values.
$\delta^{13}C$ values of mangrove yellow leaves were significantly lower (-27.8±0.2‰) than those of the
MPB (-27.1±0.05‰) (t = -3.8, p<0.01, Fig. 2c). In contrast, the ratio of C/N for mangrove
yellow leaves (95±1.8) was significantly higher than that for the MPB (16.3±0.3) (t = 44.307,
p<<0.001, Fig. 2d). $\delta^{13}C$ of benthos-respired $CO_2$ was also significantly higher when crabs fed





on MPB than when they were fasted for 2 days (-25.7±0.4‰, P<0.05) and 3 days (-25.8±0.4‰,
P<0.05). Further, $\delta^{13}C$ of $CO_2$ respired by detritivorous crabs on collection (-24.0±0.6‰) was
not significantly different from that of crab muscles (-22.6±0.6‰, P>0.05). Also no significant
differences were found between other comparisons.

**3.2 Variation of $CO_2$ production with benthos size and/or feeding regimes**
$CO_2$ production was related to consumer size and/or feeding regimes. There was a significant
difference in $CO_2$ production of crabs among different feeding regimes (ANOVA, F=2.9,
P<0.05, Fig. 4). In particular, $CO_2$ production of crabs was significantly lower when they fed on
MPB (0.13±0.02 mmol $g^{-1}$ $day^{-1}$) than on collection (0.31±0.03 mmol $g^{-1}$ $day^{-1}$, P<0.05) or fasted
for one day (0.3±0.05 mmol $g^{-1}$ $day^{-1}$, P<0.05). No significant differences were found for other
comparisons. There were significant relationships between $CO_2$ production of crabs and carapace
width ($R^2$=0.73, P<0.001, Fig. 5a), and between $CO_2$ production of crabs and carapace length
($R^2$=0.61, P<0.01, Fig. 5b). Significant differences were found in the ratios of carapace length
and width between ocypodid crabs (0.59±0.002) and sesarmid crabs (0.79±0.025) (t test, t=-7.9,
P<0.01), which application of the above relationships for intraspecific comparison may depend
on the size measurement used. The $CO_2$ production of the crabs was 0.45±0.05 mmol $g^{-1}$ $day^{-1}$ at
the average dry weight of 0.95g (0.12±0.01 mmol g wet $wt^{-1}$ $day^{-1}$). Similarly, there was a
significant relationship between $CO_2$ production of gastropods and shell length ($R^2$=0.58,
P<0.05, Fig. 5c). The $CO_2$ production of the gastropods was 0.014±0.003 mmol $g^{-1}$ $day^{-1}$ at the
average dry weight of 2.09g (0.045g without shell). No significant relationships were found
between $CO_2$ production and crab or gastropod weight (P>0.05).



**4 Discussion**

**4.1 Differences in $\delta^{13}C$ of intertidal benthos due to different food categories**

Our data suggest that $\delta^{13}C$ of benthos-respired $CO_2$ can be used to infer the categories of benthos foods being used. The $\delta^{13}C$ of benthos-respired $CO_2$ in the gastropod group was higher than that of the crab group, and that of the deposit-feeding group was higher than the detritivorous group. These different patterns may reflect different food categories of the mangrove benthos. In our laboratory experiment, the gastropods investigated mainly forage on the MPB, while the crabs may use both the MPB and mangrove leaf litter depending on their feeding habit. Yellow leaves of mangrove species in Ting Kok (where leaves and sediments were collected) have $\delta^{13}C$ at -27.8±0.16‰ and C/N at 95±1.8, while the corresponding values for MPB were -27.1±0.05‰ and 16.3±0.3. A C/N ratio of < 20 is generally required for sustainable animal nutrition (Russell-Hunter, 1970). Thus, crabs may need to use the MPB or other N-rich sources to meet their nutritional requirement to supplement the low-N leaf diets. The isotopic fractionation ($\Delta^{13}C$) is 3.2‰ and 3.9‰ between $\delta^{13}C$ of crab (-23.9±0.4‰) respired $CO_2$ and $\delta^{13}C$ of MPB and mangrove yellow leaves respectively, while it is 9.6‰ between $\delta^{13}C$ of gastropod (-17.5±1.3‰) respired $CO_2$ and $\delta^{13}C$ of the MPB (Table 1). Similarly, $\Delta^{13}C$ is 7.3‰ between $\delta^{13}C$ of deposit-feeder (including ocypodid and varunid crabs and gastropods) respired $CO_2$ (-19.8±0.8‰) and $\delta^{13}C$ of the MPB and 3.1‰ between $\delta^{13}C$ of detritivore (including sesarmid crabs) respired $CO_2$ (-24.7±0.3‰) and $\delta^{13}C$ of yellow mangrove leaves. The lack of significant difference between $\delta^{13}C$ of $CO_2$ respired by detritivorous crabs on collection and that of crab muscles is consistent with findings on terrestrial herbivores (Engel et al. 2009).

**4.2 Differences in $\delta^{13}C$ of intertidal benthos due to the feeding regime**



The significant differences in $\delta^{13}C$ of crab-respired $CO_2$ among different feeding regimes may be
attributed to fuel-switching and also to food categories consumed. Proteins, lipids and
carbohydrates are the three major classes of metabolic fuels (McCue & Welch, 2016). The long-
standing paradigm lies in that fasting animals pass through three sequential physiological phases
whereby they predominantly oxidize endogenous fuels in the sequence of carbohydrates,
followed by lipids, and then proteins (Caloin, 2004; Castellini & Rea, 1992). $\delta^{13}C$ of crab-
respired $CO_2$ was not significantly different among the MPB diet, on collection or starved for
one day, but was significantly higher compared to that of crabs starved for two or three days. In
our experiment, crabs may metabolise carbohydrates and/or mixed with lipids when they were
just collected or fasted for one day, similar to the small animals in another study (Carleton et al.,
2004). After fasting for two or three days, crabs started to consume lipids, which have $\delta^{13}C$
values 0.5-8‰ lower than those of carbohydrates (DeNiro & Epstein, 1977; McCue & Welch,
2016; Stott et al., 1997). This can explain the decline (but non-significant) in $\delta^{13}C$ of crab-
respired $CO_2$ during the fasting periods. After fasted for three days, $\delta^{13}C$ of crab-respired $CO_2$ (-
26.6±0.3‰) was about 1.2‰ higher than when they fed on yellow *Kandelia obovata* leaves.
Similarly, $\delta^{13}C$ of crab-respired $CO_2$ (-23.8±0.4‰) was about 3.3‰ higher than the diet when
they fed on MPB after starvation. These results are supported by the higher $\delta^{13}C_{breath}$ values than
$\delta^{13}C_{diet}$ (0.8-3.1‰) for different animals including steers, pigs and rabbit (Passey et al., 2005).

**4.3 Differences in benthos $CO_2$ production due to body size and feeding regime**
The significant relationships between benthos $CO_2$ production and body size suggest that animal
size distribution should be considered when estimating the contribution of the benthos to $CO_2$
emission rates from the sediment surface. Crab $CO_2$ production (0.12 mmol $g^{-1}$ wet wt $day^{-1}$) for



the sesarmid crabs in our study falls within the reported range of 0.05-0.15 mmol $g^{-1}$ wet wt $day^{-1}$
for ocypodid crabs (Kristensen et al., 2008; Penha-Lopes et al., 2010). Average gastropod $CO_2$
production in Penha-Lopes et al. (2010) (0.011 mmol $g^{-1}$ $day^{-1}$) was similar to that in our study
(0.014 mmol $g^{-1}$ $day^{-1}$).
We established the relationships between benthos $CO_2$ production and body size (carapace
width or length). They can be used for different purposes. Particularly, when estimating crab
$CO_2$ production related to crab burrow size (i.e. the diameter of burrow openings) (e.g. Cameron
et al. (2019)), the relationship between $CO_2$ production and carapace length rather than carapace
width should be used as crabs walk side-ways. However, carapace length is not as good a
measurement of animal body size as carapace width when different crab families are involved.
For example, our data suggest that ocypodid crabs had significantly smaller carapace
length/width ratios (0.59±0.002) than those of sesarmid crabs (0.79±0.025).
The significant impact of feeding regimes on $CO_2$ production of sesarmid crabs may be a
result of the differences in carbon content of the diet and endogenic fuel. $CO_2$ is a waste product
of oxidising reduced carbon compounds during crab metabolism. When sesarmid crabs were just
collected from the field, their main carbon source is mangrove leaves, which were found to have
a higher carbon content than the MPB (45.6% vs. 30.6%) (Bui & Lee, 2014). This could account
for the significant differences in $CO_2$ production of crabs on collection and while feeding on the
MPB. As small animals, when fasted for one day, our collected crabs may consume a mixture of
recent diet (mangrove leaves) and stored carbohydrates, which are easier to decompose than the
structural carbon-rich mangrove leaves and other organic compounds, e.g. lipids and proteins.
However, when they were fasted for two or three days, their energy source may shift to lipids
and proteins, which are more difficult to be metabolised to $CO_2$ than carbohydrates. This could



account for the significantly higher $CO_2$ production of crabs when they were fasted for one day
but not more days than they fed on MPB. $CO_2$ production of *Carollia perspicilata* was also
reported to generally decline over time after feeding (Welch Jr et al., 2016), corroborating with
our findings.

**5 Conclusions and Implications**
Our study tested the hypothesis that intertidal benthos breathe what they eat. This hypothesis is
supported by the significantly higher $\delta^{13}C$ of deposit feeders-respired $CO_2$ than that of
detritovore-respired $CO_2$, and the significant difference in $CO_2$ production under different
feeding regimes due to different carbon content in diet or decomposability of carbon compounds
in fuel.
Our study supports the notion that $\delta^{13}C$ of benthos-respired $CO_2$ can be used to differentiate
food categories of the benthos, as has been demonstrated in mammals and other animals. It may
be further applied to reveal the contribution of different food sources to their diet over short and
long periods when combined with (compound-specific) stable isotope analyses of animal tissues
and diets. For partitioning the food sources of intertidal animals, past studies combining stable
isotope analyses of animal tissues and their diet have obvious limitations because they can only
figure out their integrated food over long periods. However, intertidal benthos (e.g. crabs) may
consume endogenous fuels such as lipids which are lower in $^{13}C$ (Stott et al., 1997) when they
are inactive without feeding. There are also differences among endogenous fuels. For example,
the difference in $\delta^{13}C$ of synthesised lipids and carbohydrates are assumed to be -3‰ (Carleton
et al., 2004). Nevertheless, few studies can exclude the influence of endogenous fuels on the
apparent food sources of intertidal benthos when using stable isotope analyses of animal tissues.



Our study also reiterates the necessity of integrating the influence of benthos feeding regime
and size into estimating the contribution of intertidal benthos to $CO_2$ emission rates from the
sediment surface. To date, a few studies have established benthos $CO_2$ production related to crab
mass but are limited to a few families, e.g. ocypodid crabs (Penha-Lopes et al., 2010) and
sesarmid crabs in this study. The former study established a significant polynomial relationship
between crab $CO_2$ production and body mass for ocypodid crabs, while we found a significant
linear relationship between crab $CO_2$ production and size (i.e. crab carapace width/length and
gastropod shell length) for sesarmid crabs. It highlights that the relationships between benthos
$CO_2$ production and size indicators (i.e. weight or size) may be different depending on the
benthos taxa or families. Therefore, caution should be exercised in estimating benthos $CO_2$
production from size indicators based on relationships developed for other benthos taxa.
Moreover, our study demonstrates the importance of considering the activity status of the
animals when estimating benthos $CO_2$ production; in particular, a significant consideration for
up-scaling estimates on the contribution of benthos $CO_2$ production to system sediment $CO_2$
emission rates. The same benthic animals may forage alternatively on MPB and mangrove
litter/detritus, as well as switch between active-feeding and fasting over long periods.

**Code availability**
Computer codes are available upon request sent to Xiaoguang Ouyang.

**Authors' contributions**



XO and SYL designed the study. XO conducted field survey, laboratory experiment and data
analysis, and wrote the manuscript revised by SYL. CYL contributed to data acquisition and
analysis.

**Competing interests**
The authors declare that they have no conflict of interest.

**Acknowledgements**
We thank Yan Ping Loo and Fen Guo for their assistance in collecting the crabs. Ms Kit Sum
Leung (The University of Hong Kong) is thanked for conducting the stable isotope analysis.
Professor Brian Fry (Griffith University, Australia) is acknowledged for his constructive advice
on the initial manuscript drafts. Xiaoguang Ouyang was supported by a Postdoctoral Fellowship
at The Chinese University of Hong Kong.

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





**Figure legends**

**Fig.1** A conceptual model showing the process of measuring $CO_2$ production and $\delta^{13}C$ of respired $CO_2$ by CRDS: the closed container, a sesarmid crab, the syringe used to collect gas from the container, a stop-cock, and CRDS. CRDS denotes cavity ring-down spectroscopy.

**Fig. 2** The variation of $\delta^{13}C$ of benthos-respired $CO_2$ with different benthos taxa (a), dominant feeding habits (b) and diets (c), as well as C/N of diets (d). Bars with different letters have significantly different values.

**Fig. 3** The variation of $\delta^{13}C$ of crab-respired $CO_2$ with different feeding regimes. Bars with different letters are significantly different.

**Fig. 4** The variation of the rate of crab $CO_2$ production with different feeding regimes. MPB denotes microphytobenthos. Bars with different letters have significantly different $CO_2$ production.

**Fig. 5** The relationships between crab $CO_2$ production and carapace width (a), and carapace length (b), as well as between gastropod $CO_2$ production and shell length (c).



Fig. 1

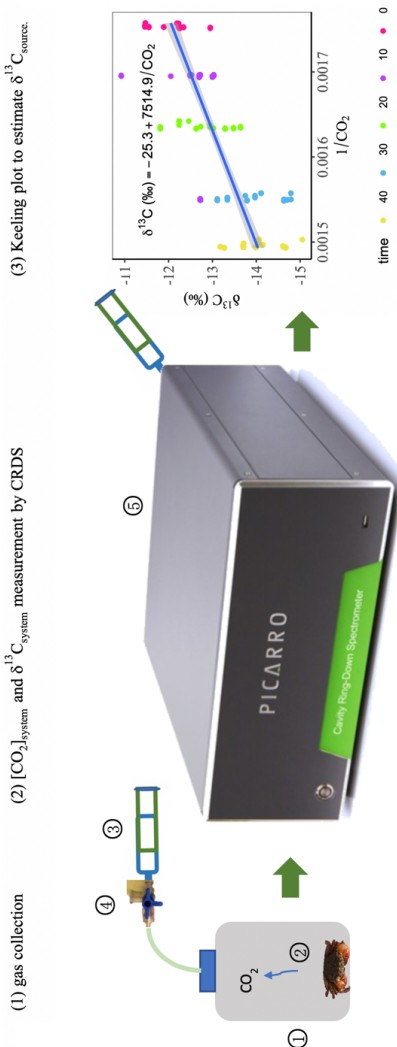





Fig. 2
a)                                                    b)

c)                                                    d)







Fig. 3

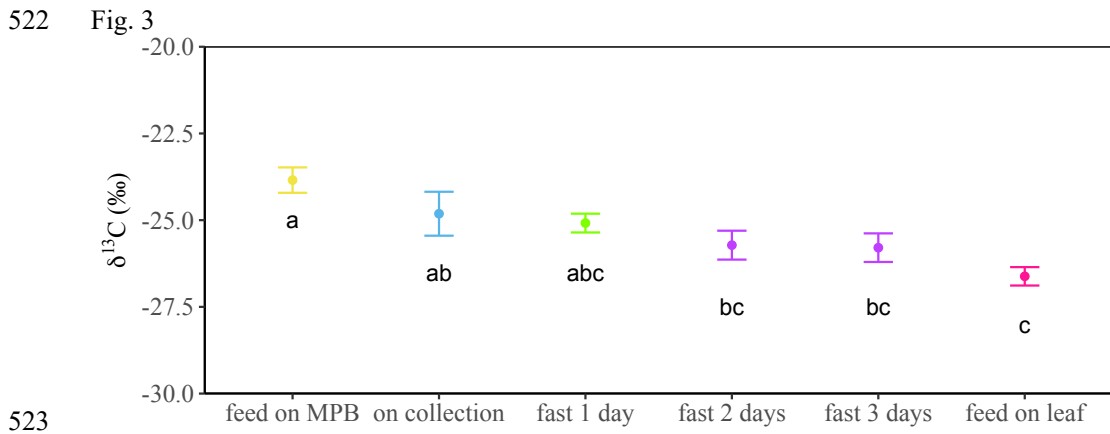




Fig. 4

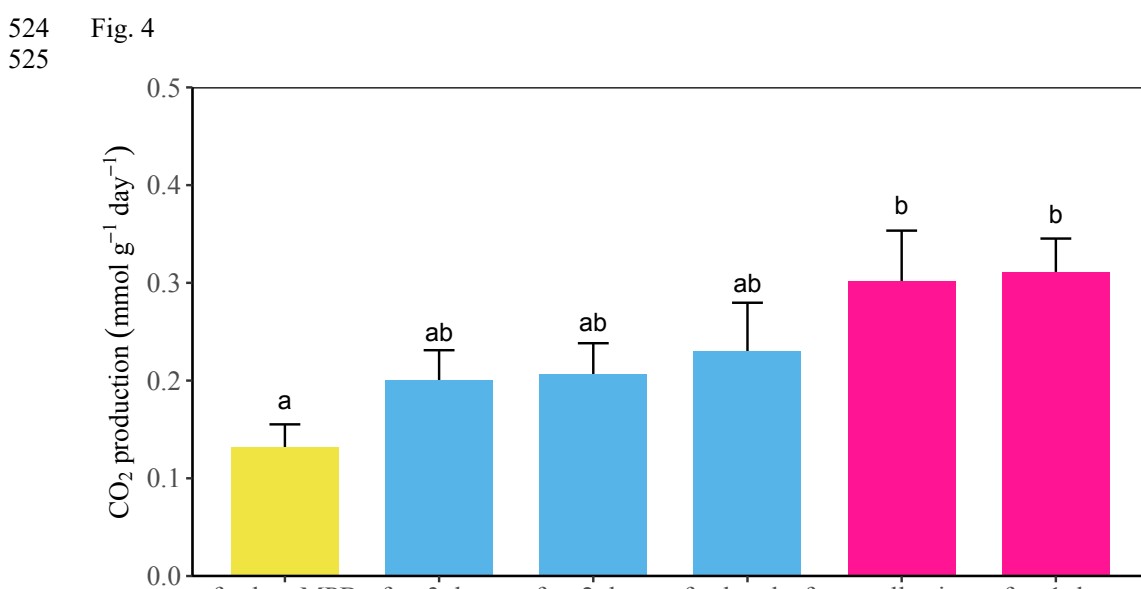



Fig. 5
a)






b)





c)













Table 1 Isotopic fractionation between mangrove animal respired $CO_2$ and their diets. MPB
denotes microphytobenthos.

| Benthos taxa | | Diet | | Isotopic fractionation | |
|---|---|---|---|---|---|
| $\delta^{13}C$ of crab respired $CO_2$ | $\delta^{13}C$ of gastropod respired $CO_2$ | $\delta^{13}C$ of MPB | $\delta^{13}C$ of yellow leaves | $\Delta^{13}C_{benthos-MPB}$ | $\Delta^{13}C_{benthos-leaves}$ |
| -23.9±0.4‰ | -17.5±1.3‰ | -27.1±0.05‰ | -27.8±0.2‰ | 3.2‰[a], 9.6‰[b] | 3.9‰[a] |
| Feeding habit | | Diet | | Isotopic fractionation | |
| $\delta^{13}C$ of deposit-feeder respired $CO_2$ | $\delta^{13}C$ of detritivore respired $CO_2$ | $\delta^{13}C$ of MPB | $\delta^{13}C$ of yellow leaves | $\Delta^{13}C_{deposit-feeder-MPB}$ | $\Delta^{13}C_{detritivore-leaves}$ |
| -19.8±0.8‰ | -24.7±0.3‰ | -27.1±0.05‰ | -27.8±0.2‰ | 7.3‰ | 3.1‰ |

[a] crab, [b] gastropod