# Peer review of "Do marine benthos breathe what they eat?"

_Biogeosciences, 2020_

## Referee Comment (RC1) · Anonymous Referee #1 · 17 Dec 2020

In this manuscript, Ouyang et al. present an interesting method to measure the $\delta^{13}$C-$CO_2$ production of various intertidal species using Cavity-Ring Down Spectroscopy (CRDS). The authors aim to link the $\delta^{13}$C-$CO_2$ production to the food sources of these species and advocate the concept "You breathe what you eat".

Identifying food sources from field-collected organisms is an important and timely topic in food web research, so I think any attempt to add an useful technique to the toolbox of food web researchers is welcome. The authors show that there is a lot of potential in obtaining precise measurements of $CO_2$ and $\delta^{13}$C-$CO_2$ production that can be achieved with the CRDS for intertidal fauna. Unfortunately, I also think that this study does not live up to this potential. The manuscript describes a series of four, only loosely connected, experiments that are poorly described and have only marginal scientific

significance. Most importantly however, the advocated concept "You breathe what you eat" is not confirmed by the experiments nor it is clear why the labor-intensive and expensive CRDS is superior to the traditional $\delta^{13}$C-tissue analysis. I detail my major concerns below.

1. The concept "You are what you eat" is simple: It assumes that the isotope composition of an organism is a simple mixture of its food sources (in case of $\delta^{13}$C) or a fixed fractionation factor heavier (in case of $\delta^{15}$N). This means that any researcher can 'simply' collect a large number of organisms from the field, analyse them for their isotope signature and reconstruct its diet, making it a very powerful technique. Of course, there are several potential caveats and problems. The concept "You breathe what you eat" in contrast is methodologically significantly more complex and expensive. If it would resolve some of the caveats associated with the "you are what you eat" concept, then it would be a very welcome addition. Table 1 shows however that organisms **do not** breathe what they eat. Instead, there is a clear, species- **and** diet-specific fractionation factor between diet consumed and $\delta^{13}$C-$CO_2$ produced. So why bother going through all the hassle of this more complex method? In addition, the authors do not show or discuss how the classical tissue isotope analyses (samples are measured though, see line 155-156) compares to the produced $\delta^{13}$C-$CO_2$ isotope values.

2. I am not sure how to interpret the experimental design from lines 132-139, but it seems that fauna were kept in a 800-mL container that was covered with punctured aluminium foil. Air samples (30-mL) from the container were taken at several sequential time points. I may be wrong, but from this I understand that each sample extraction will 'suck in' ambient air into the container, which will dilute the produced $CO_2$ in the container. How did the authors correct for this or did I misunderstand something here?

3. Overall, the experimental procedures are not well described. I found it difficult to reconstruct exactly how the different experiments were conducted, why some organisms were starved, whether starved was considered similar to dormant (not the same in my opinion, but see line 128-129), how many replicates were done etc. In addition,

the experiments are 'numbered' in the Materials and Methods section for clarity, but this numbering is not followed in the Results section, so linking results to methods is cumbersome.

4. $CH_4$ measurements were conducted (line 160) but the data are not presented.

5. Also $CO_2$ production of differently sized organisms is measured, which gives the rather trivial (yet very useful for doing respiration budget studies) relation of increasing $CO_2$ production with body size. Many authors have used and use the cheaper and easier method of continuous measurements of oxygen concentration of submerged organisms in an incubation chamber. I would really like to read what the complex CRDS measurements offer **in addition** to providing straightforward respiration data.

---

## Referee Comment (RC2) · Anonymous Referee #2 · 5 Jan 2021

This paper aims to determine if "you breathe what you eat", specifically, whether the d13C value of carbon dioxide respired by benthic fauna (crabs and gastropods) from a mangrove forest reflects the d13C of the dietary source. The authors used four separate experiments to determine how d13C of respired carbon is affected by 1) taxon (crabs vs gastropods), 2) feeding mode (detritivores vs deposit-feeders), and 3) food source (MPB vs senescent mangrove leaves, and 4) how the quantity of respired $CO_2$ is affected by animal size. The flux and d13C of respired $CO_2$ was determined through incubation of fauna within sealed chambers, with $CO_2$ sampled periodically via syringe, and analysed using a Picarro CRDS.

Although there are some very clear differences and trends in the data collected, I unfortunately remain unconvinced that this study represents a useful advance in isotope

techniques and I remain uncertain of the conclusions. The manuscript should be revised to improve readability through correction of grammatical errors. However, it is most important that the manuscript be revised to more clearly and thoroughly outline the basis for the study and the hypothesis of the various experiments, and to clearly explain how the outcomes of the study advance the field and can be practically applied elsewhere. My main concerns are as follows:

1) As per the comment above, the justifications, aims, and hypotheses of each of the experiments should be clearly outlined.

2) Given that d13C values for the leaf and MPB are similar, and 13C fractionation between diet and respired CO2 is apparently ~3-10 per mil (Table 1) and presumably somewhat variable, it seems to me that it would be impossible to distinguish leaves and MPB as potential dietary sources on the basis of breath analysis. Even for sources with more distinct d13C values (e.g., C3 and C4 plants) it would be difficult to determine diet, given the large variability in fractionation of d13C-CO2resp vs d13C-diet based on taxa, diet, and feeding mode (based on data in Table 1). It appears that very specific data would need to be collected through targeted experiments before applying the proposed method, which seems to make application of the proposed technique too complex to be practical.

3) Assuming it 'works', the advantage of using the proposed technique is unclear. The d13C of respired CO2 (d13C-CO2resp) has been used to determine diets of higher order consumers where it is not feasible to collect tissue samples for 'traditional' isotope analysis using an EA-IRMS. However, this is typically not an issue for intertidal invertebrates, and CRDS is still typically less accessible than EA-IRMS analysis. Where might the proposed method be of use? Given the rapid shift in d13C-CO2resp it is likely that the CO2 represents very recent diet, and could be used in combination with tissue sampling to determine diet shifts. Could the authors indicate where this might be useful? Furthermore, how might it be possible to distinguish between a shift in recent diet vs a shift to use of stored carbon (e.g. lipids)? How is this distinct from simply

analysing tissues with different turnover rates to look at recent changes in diet? It could also be noted that the method does not rely on use of a CRDS, but CO2 samples could also be collected into sealed vials and submitted for analysis via a gasbench/GC-IRMS where a CRDS is unavailable.

4) The title of the paper "Do marine benthos breathe what they eat?" does not reflect the content of the manuscript in its entirety. There is considerable focus on whether marine benthos breathe more when they are larger (CO2 flux vs size relationships). With regard to this focus, it is not clear how an understanding of CO2 production vs size is of practical use. The authors mention the potential incorporation of this relationship when determining the contribution of fauna to CO2 effluxes from mangrove forests, but this would presumably also rely on some understanding of population structure and/or size distribution of benthic taxa. This should be outlined to make clear why this should be of interest.

5) The implications of the large differences in fractionation with taxa, diet, and feeding mode are not fully discussed. Fauna may breathe what they eat, but how can we determine what they eat based on what they breathe?

6) What is the potential impact of confinement and the conditions of the chambers (no sediment, no burrow) on respiration (both in terms of quantity of C respired, and its source (e.g. lipids vs carbohydrates vs proteins))

My more specific comments are as follows:

The methods appear quite straightforward, but some additional information should be provided. Specifically:

1) What was the potential for dilution of gas within each chamber with air entering through the hole in the foil and/or around the edge of the foil (was this sealed in place)?

2) How many animals of each taxa were used in each experiment? How many animals were in each chamber?

3) Ln 143: Presumably the MPB fed to the fauna were obtained through sieve and spin, otherwise what was the source of MPB? If MPB were from sieve and spin, was the MPB confirmed to still be living? How was the MPB provided, given that the final step was concentration on a filter?

4) Was experiment 4 run separately to the other experiments? Were animal sizes standardised in the other experiments?

5) CH4 analysis is mentioned, but no data is presented.

6) How many leaf and MPB samples were analysed? What area were these collected from? Where were these collected vs where were animals collected?

7) Crab tissues were apparently analysed after incubation, but this data is not presented. It would be interesting to see how d13C values for muscle and other tissues compared to d13C-CO2resp. Were gastropods analysed for tissue d13C?

8) Confirm sampling times: 20 minutes after incubation began then every 10 minutes over 50 additional minutes?

Throughout the ms, replace lower/higher with 13C-depleted/13C-enriched (or similar)

It would make reading easier if the authors replaced "d13C of deposit feeders respired CO2", which is grammatically awkward, with "d13C-CO2resp for deposition feeders". This abbreviation (or similar) could be used throughout the ms to improve readability.

Ln 14-16 and elsewhere: It is not immediately clear what is meant by "feeding regime"

Ln 26: "on field collection" – it is unclear what this means without having read the remainder of the paper

Ln 39: Remove 'past'

Ln 47 onwards: Rework. The focus here on predators seems at odds with the focus of this manuscript on detritivores and deposit feeders.

Ln 48: Provide references for previous use of d13C-CO2 techniques

Ln 75-76: Unclear.

Ln 125-130: Some of this information would be better placed in the introduction

Ln 126: Apparently crabs and gastropods were compared, but it seems likely there could be just as much different between the crab groups (ocypodids vs sesarmids) as between crabs and gastropods.

Ln 140: remove one mention of "collected"

Ln 143: Unclear what is meant by "gas samples were collected separately".

Ln 153-154: Presumably the samples were homogenised. Were the crabs dissected to remove tissues first?

Ln 197&Ln 200: Specify what the groups were (e.g. crabs vs gastropods, or different crab groups vs gastropods?)

Ln 218-219 seems repeated in ln 219-220.

Ln 227-230: Somewhat misleading. The pattern is consistent, but the magnitude is different.

Ln 245 (and elsewhere): reduce repetition, "CO2 production significantly increased with carapace length" (remove "there was a significant relationship. . ."_

Ln 249-250: Sentence is unclear.

Ln 259: It is not clear what is meant by 'categories'.

Section 4.1: The point of this section is unclear. It seems to mainly repeat the results, with no new inferences apparent. Can the authors specifically explain how d13C-CO2resp can indicate dietary sources in some useful way?

Ln 294: The relevance of the similarity of d13C for fasting crabs and those fed on

leaves is unclear

Ln 297: Interesting that fractionation of d13C-CO2resp vs d13C-diet for crabs is similar to the range here, but gastropods seem to have far greater fractionation. Why? Where the diet of crabs was switched, is it possible that they were still using stored C (e.g. lipids), and this would have diluted the d13C value of the respired CO2 and affected results seen here?

Ln 327: Specify that C. perspicillata is a bat, and check the spelling of the species name.

Ln 334: Check spelling of detritivore

Figure 1: There is one purple point in among the blue points – is this an error?

Figure 4: The order of the bars (left to right) appears non-intuitive and does not match Figure 3. Also, the colours in this and other figures is unnecessary.

Table 1: Is it possible to provide an error estimate for the fractionation values? E.g. d13C of individual animals – d13C of diet?

---

## Author Comment (AC1) · 27 Jan 2021

In this manuscript, Ouyang et al. present an interesting method to measure the $\delta$13C-CO2 production of various intertidal species using Cavity-Ring Down Spectroscopy (CRDS). The authors aim to link the $\delta$13C-CO2 production to the food sources of these species and advocate the concept "You breathe what you eat".

Response: We thank the reviewer for the compliments on our manuscript.

Identifying food sources from field-collected organisms is an important and timely topic in food web research, so I think any attempt to add an useful technique to the toolbox of food web researchers is welcome. The authors show that there is a lot of potential in obtaining precise measurements of CO2 and $\delta$13C-CO2 production that can be

achieved with the CRDS for intertidal fauna. Unfortunately, I also think that this study does not live up to this potential. The manuscript describes a series of four, only loosely connected, experiments that are poorly described and have only marginal scientific significance. Most importantly however, the advocated concept "You breathe what you eat" is not confirmed by the experiments nor it is clear why the labor-intensive and expensive CRDS is superior to the traditional $\delta$13C-tissue analysis. I detail my major concerns below.

Response: We shall rearrange the structure of the manuscript to make the linkage of the four experiments be more apparent. Specifically, in the Results section, we will arrange the result of each experiment clearly corresponding to that described in the Materials and Methods section. We do not agree that CRDS is labor-intensive and expensive. For measurement of gas samples, CRDS offers a quick and inexpensive alternative to traditional IRMS, as there is less need for calibration. The standalone G2201i used in our study is less expensive compared with most IRMS.

1. The concept "You are what you eat" is simple: It assumes that the isotope composition of an organism is a simple mixture of its food sources (in case of $\delta$13C) or a fixed fractionation factor heavier (in case of $\delta$15N). This means that any researcher can 'simply' collect a large number of organisms from the field, analyse them for their isotope signature and reconstruct its diet, making it a very powerful technique. Of course, there are several potential caveats and problems. The concept "You breathe what you eat" in contrast is methodologically significantly more complex and expensive. If it would resolve some of the caveats associated with the "you are what you eat" concept, then it would be a very welcome addition. Table 1 shows however that organisms do not breathe what they eat. Instead, there is a clear, species- and diet-specific fractionation factor between diet consumed and $\delta$13C-CO2 produced. So why bother going through all the hassle of this more complex method? In addition, the authors do not show or discuss how the classical tissue isotope analyses (samples are measured though, see line 155-156) compares to the produced $\delta$13C-CO2 isotope values.

Response: We acknowledge that there is fractionation between diet consumed and $\delta$13C of benthos respired CO2 which also exists between diet consumed and $\delta$13C of an organism. The advantages of our method lie in the following aspects: (1) it can provide information about both the most recently consumed diet and the integrated diet over longer periods while the classic tissue isotope analysis only tracks the integrated diet over time. (2) breath $\delta$13C can be repeatedly measured non-destructively for the same animals and thus can track the changes in its food sources while animals must be sacrificed for the classic tissue isotope analysis which cannot track the change in food sources for the same animal. Our method is useful since some marine crabs remain dormant most of the time with a short active period (e.g. 90 days, Katz 1980). Our experiment has monitored the changes in $\delta$13C of benthos respired CO2 and CO2 production when they are fasted or fed on leaf litter/microphytobenthos to reflect their active and dormant status. Some species of aquatic migratory species occupy intertidal habitats during specific seasons of the year. Our experiment has shown the changes in $\delta$13C of benthos respired CO2 and CO2 production under different feeding regime to reflect their changes in food during migration. CRDS can be replaced by alternatives which can fulfil the same objective. For example, gas samples can be collected in sealed vials and analysed via a gasbench/gas chromatography coupled with Isotope Ratio Mass Spectrometer. We have shown the comparison of $\delta$13C of CO2 respired by crabs with classic tissue isotope analysis but found no significant difference between them (Line 235-7). We shall revise the text to strengthen the description on the advantage and repeatability of our methods.

2. I am not sure how to interpret the experimental design from lines 132-139, but it seems that fauna were kept in a 800-mL container that was covered with punctured aluminium foil. Air samples (30-mL) from the container were taken at several sequential time points. I may be wrong, but from this I understand that each sample extraction will 'suck in' ambient air into the container, which will dilute the produced CO2 in the container. How did the authors correct for this or did I misunderstand something here?

Response: The small hole (diameter: 2mm) on the lid is designed to keep the pressure balance between the inside and outside of the containers but will not result in abrupt air exchange. The small hole used for ventilation has been demonstrated in the previous studies (e.g. Carleton et al. 2004).

3. Overall, the experimental procedures are not well described. I found it difficult to reconstruct exactly how the different experiments were conducted, why some organisms were starved, whether starved was considered similar to dormant (not the same in my opinion, but see line 128-129), how many replicates were done etc. In addition, the experiments are 'numbered' in the Materials and Methods section for clarity, but this numbering is not followed in the Results section, so linking results to methods is cumbersome.

Response: We thank the reviewer for reminding us of describing the experiments and results more clearly for repeatability. Some animals were starved to simulate the dormant status under which they would not feed. There are 15 replicates for each group in experiment 1, 10 replicates for each group in experiment (2), 15 replicates for each group in experiment (3). We'll supplement this information in the revised version and rearrange the results, corresponding to the numbering in the Materials and Methods section. See our response to the reviewer's overall comments.

4. CH4 measurements were conducted (line 160) but the data are not presented.

Response: We have responded to this query in response to the specific comment 5) of reviewer #2.

5. Also CO2 production of differently sized organisms is measured, which gives the rather trivial (yet very useful for doing respiration budget studies) relation of increasing CO2 production with body size. Many authors have used and use the cheaper and easier method of continuous measurements of oxygen concentration of submerged organisms in an incubation chamber. I would really like to read what the complex CRDS measurements offer in addition to providing straightforward respiration data.

Response: The measurement of $CO_2$ production of different organisms is useful if combined with population structure and/or size distribution of the benthos to determine how much marine fauna contribute to $CO_2$ effluxes from mangrove forests. Our method measures $CO_2$ production from organisms in mangroves, which are emersed during low tides. There are alternatives for our instrument as indicated in our response to the reviewer's major concern 1. If the purpose is just to measure $CO_2$ production of organisms, cheaper options (e.g. infrared gas analysers) can be applied. We shall state these points in the revised version.

References Carleton, S. A., Wolf, B. O., & Del Rio, C. M. (2004). Keeling plots for hummingbirds: a method to estimate carbon isotope ratios of respired $CO_2$ in small vertebrates. Oecologia, 141(1), 1-6. Katz, L. C. (1980). Effects of burrowing by the fiddler crab, Uca pugnax (Smith). Estuarine and Coastal Marine Science, 11(2), 233-237.

---

## Author Comment (AC2) · 27 Jan 2021

This paper aims to determine if "you breathe what you eat", specifically, whether the d13C value of carbon dioxide respired by benthic fauna (crabs and gastropods) from a mangrove forest reflects the d13C of the dietary source. The authors used four separate experiments to determine how d13C of respired carbon is affected by 1) taxon (crabs vs gastropods), 2) feeding mode (detritivores vs deposit-feeders), and 3) food source (MPB vs senescent mangrove leaves, and 4) how the quantity of respired $CO_2$ is affected by animal size. The flux and d13C of respired $CO_2$ was determined through incubation of fauna within sealed chambers, with $CO_2$ sampled periodically via syringe, and analysed using a Picarro CRDS. Although there are some very clear differences and trends in the data collected, I un- fortunately remain unconvinced that this study

represents a useful advance in isotope techniques and I remain uncertain of the conclusions. The manuscript should be revised to improve readability through correction of grammatical errors. However, it is most important that the manuscript be revised to more clearly and thoroughly outline the basis for the study and the hypothesis of the various experiments, and to clearly explain how the outcomes of the study advance the field and can be practically applied elsewhere.

Response: We will revise the ms to correct some grammatical errors, such as replacing $\delta$13C of deposit feeders respired CO2 with $\delta$13C-CO2 respiration for deposit feeders throughout the ms. We will clearly present the justification, aim and hypothesis of each experiment. We will clearly show the advantage of our method in comparison with classical tissue isotope studies, and specify the alternative for CRDS so that the concept 'You Breathe What You Eat' can be applied elsewhere. Specifically, we have clarified how we will improve the ms in the following responses to the reviewer's concerns and specific comments.

My main concerns are as follows: 1) As per the comment above, the justifications, aims, and hypotheses of each of the experiments should be clearly outlined.

Response: We aim to put forward the novel notion 'You Breathe What You Eat', which adds to the well-known paradigm of 'You Are What You Eat' in food web studies. Specifically, whether the 13C-CO2 respiration for marine benthos reflects their taxonomic background, size and dietary source. Our specific hypotheses include (1) 13C-CO2 respiration is dependent on the taxonomic background of benthic consumers; (2) 13C-CO2 respiration reflects their feeding habit; (3) 13C-CO2 respiration and CO2 production is influenced by the feeding regime; and (4) benthos CO2 production is dependent on their size. We shall rearrange the Results section to make it correspond to each experiment described in the Materials and Methods section and hypotheses outlined in the Introduction section, to improve correspondence between results of each experiment and the relevant hypothesis.

2) Given that d13C values for the leaf and MPB are similar, and 13C fractionation between diet and respired CO2 is apparently âĹij3-10 per mil (Table 1) and presumably somewhat variable, it seems to me that it would be impossible to distinguish leaves and MPB as potential dietary sources on the basis of breath analysis. Even for sources with more distinct d13C values (e.g., C3 and C4 plants) it would be difficult to determine diet, given the large variability in fractionation of d13C-CO2resp vs d13C-diet based on taxa, diet, and feeding mode (based on data in Table 1). It appears that very specific data would need to be collected through targeted experiments before applying the proposed method, which seems to make application of the proposed technique too complex to be practical.

Response: In Fig. 1, we have shown that there is a significant difference between ïĄd'13C of mangrove leaf and MPB (t test, P<0.01). Our results show the apparent fractionation between ïĄd'13C-CO2 respiration for benthos and their diets (3.1-9.6 ‰ resulting from different benthos taxa and diets. In classical tissue isotope studies, clear fractionation differences are also demonstrated between ïĄd'13C of crabs tissues and diets (Bui and Lee 2014) and different benthic consumers (Kristensen et al. 2017). So we think taxa and diet specific isotope fractionation is common for both our approach and the classical tissue isotope approach, and does not hinder its use for identifying the food sources of benthos. The solution is to use diet and taxa specific isotope fractionation to determine benthos diets, as the fractionation value generating from a wide range of consumer-food combinations have been criticized for failing to explain specific trophic paths (Bui and Lee 2014). We shall revise the text to clearly show these points.

3) Assuming it 'works', the advantage of using the proposed technique is unclear. The d13C of respired CO2 (d13C-CO2resp) has been used to determine diets of higher order consumers where it is not feasible to collect tissue samples for 'traditional' isotope analysis using an EA-IRMS. However, this is typically not an issue for intertidal invertebrates, and CRDS is still typically less accessible than EA-IRMS analysis. Where

might the proposed method be of use? Given the rapid shift in d13C-CO2resp it is likely that the CO2 represents very recent diet, and could be used in combination with tissue sampling to determine diet shifts. Could the authors indicate where this might be useful? Furthermore, how might it be possible to distinguish between a shift in recent diet vs a shift to use of stored carbon (e.g. lipids)? How is this distinct from simply analysing tissues with different turnover rates to look at recent changes in diet? It could also be noted that the method does not rely on use of a CRDS, but CO2 samples could also be collected into sealed vials and submitted for analysis via a gasbench/GC-IRMS where a CRDS is unavailable.

Response: We thank the reviewer for suggesting gasbench/GC-IRMS as an alternative to our approach. The concept 'You Breathe What You Eat' does not rely on the use of CRDS but this equipment facilitates the measurements. The usefulness of our method lies in: (1) it can provide information about both the most recently consumed diet and the integrated diet over longer periods while the classic tissue isotope analysis only tracks the integrated diet over time; (2) breath $\delta$13C can be repeatedly measured non-destructively for the same animals and thus can track the changes in its food sources while animals must be sacrificed for the classic tissue isotope analysis which cannot track the change in food sources for the same animal. Our method is useful since some marine crabs remain dormant most of the time with a short active period (e.g. 90 days, Katz 1980). Our experiment has monitored the changes in 13C-CO2 respiration for benthos and CO2 production when they are fasted or fed on leaf litter/microphytobenthos to reflect their active and dormant status. Some species of aquatic migratory species occupy intertidal habitats during specific seasons of the year. Our experiment has shown the changes of 13C-CO2 respiration and CO2 production under different feeding regimes to reflect their changes in food during migration. When $\delta$13C-CO2 respiration for benthos is combined with tissue sampling, it might be useful for identifying the shift in recent diet versus using stored energy under starved conditions by analysing tissues with different turnover rates.

4) The title of the paper "Do marine benthos breathe what they eat?" does not reflect the content of the manuscript in its entirety. There is considerable focus on whether marine benthos breathe more when they are larger ($CO_2$ flux vs size relationships). With regard to this focus, it is not clear how an understanding of $CO_2$ production vs size is of practical use. The authors mention the potential incorporation of this relationship when determining the contribution of fauna to $CO_2$ effluxes from mangrove forests, but this would presumably also rely on some understanding of population structure and/or size distribution of benthic taxa. This should be outlined to make clear why this should be of interest.

Response: We shall highlight the significance of the relationships between $CO_2$ flux and benthos size, combined with population structure and/or size distribution of benthic taxa to determine how much marine fauna contribute to $CO_2$ effluxes from mangrove forests. The latter data on size distribution and density are, however, beyond the scope of this study.

5) The implications of the large differences in fractionation with taxa, diet, and feeding mode are not fully discussed. Fauna may breathe what they eat, but how can we determine what they eat based on what they breathe?

Response: As explained in our response to the reviewer's concern (2), we will discuss and clarify the implications of the taxa and diet specific fractionation, and the approach to determine what they eat based on what they breathe.

6) What is the potential impact of confinement and the conditions of the chambers (no sediment, no burrow) on respiration (both in terms of quantity of C respired, and its source (e.g. lipids vs carbohydrates vs proteins))

Response: In experiment (4), we only tried to determine the relationship between benthos respired $CO_2$ and their sizes. Our previous study has examined the relationship between number of crab burrows and sediment $CO_2$ flux (Ouyang et al., 2017). These studies and other related studies are useful for partitioning the contribution of marine

benthos and burrows to CO2 effluxes from mangrove forests. Even in the container, we observed that crabs and gastropods behaved normally. Without putting them in a container, it is impossible to measure their respiration. Inclusion of elements like sediment would also introduce sources of error due to sediment respiration, etc., which are obviously undesirable. The impact of the confinement is difficult to examine but we will acknowledge this limitation in the revised version.

My more specific comments are as follows: The methods appear quite straightforward, but some additional information should be provided. Specifically: 1) What was the potential for dilution of gas within each chamber with air entering through the hole in the foil and/or around the edge of the foil (was this sealed in place)?

Response: The small hole (diameter: 2mm) on the lid is designed to keep the pressure balance between the inside and outside of the containers but will not result in abrupt air exchange. The small hole used for ventilation has been demonstrated in the previous studies (e.g. Carleton et al. 2004).

2) How many animals of each taxa were used in each experiment? How many animals were in each chamber?

Response: There were 15 replicates for each taxon in experiment 1, 10 replicates for each group in experiment 2, 15 replicates for each group in experiment 3, and one animal was put in each chamber each time to measure its CO2 efflux in experiment 4. These numbers will be included in the revised version.

3) Ln 143: Presumably the MPB fed to the fauna were obtained through sieve and spin, otherwise what was the source of MPB? If MPB were from sieve and spin, was the MPB confirmed to still be living? How was the MPB provided, given that the final step was concentration on a filter?

Response: Yes, the MPB fed to the fauna were obtained via the sieve and spin method. When we checked the MPB in the top layer of the separated solution under microscope,

we can observe the motility of the MPB. The MPB was collected on pre-combusted GF/F filters, which were put in the containers and the benthos feed on the MPB on the filters.

4) Was experiment 4 run separately to the other experiments? Were animal sizes standardised in the other experiments?

Response: Yes, experiment 4 was run separately. We standardized animal sizes (line 187-8).

5) CH4 analysis is mentioned, but no data is presented.

Response: CRDS can simultaneously measure CO2 and CH4 concentrations and isotope values but CH4 concentrations are too low to build up in the container and are therefore not used in our analysis. We shall clarify this in the revised ms.

6) How many leaf and MPB samples were analysed? What area were these collected from? Where were these collected vs where were animals collected?

Response: We collected senescent leaves in one zip-lock bag (23 ×15cm), and 10 bags of surface sediments to separate the MPB. The samples were collected from Ting Kok mangroves, Hong Kong. The animals were collected from Ting Kok and Mai Po mangroves (Line 110-1).

7) Crab tissues were apparently analysed after incubation, but this data is not presented. It would be interesting to see how d13C values for muscle and other tissues compared to d13C-CO2resp. Were gastropods analysed for tissue d13C?

Response: We have shown the comparison of $\delta$13C-CO2 respiration for crabs with classic tissue isotope analysis but found no significant difference between them (Line 235-7). Gastropods were not analysed for tissue $\delta$13C.

8) Confirm sampling times: 20 minutes after incubation began then every 10 minutes over 50 additional minutes?

Response: It is 20 minutes after incubation began then every 10 minutes over 40 additional minutes. Twenty minutes after incubation is the start point (0 minutes) and the sampling sequence is 0, 10, 20, 30, 40 minutes. We shall replace 50 minutes (Line 139) with 40 minutes in the text.

Throughout the ms, replace lower/higher with 13C-depleted/13C-enriched (or similar)

Response: In our manuscript, we compare $\delta$13C-CO2 for animals of different taxa, feeding habits and regime. "Lower/higher" are the preferred terminology for describing $\delta$13C values to "depleted/enriched" (see Bond and Hobson 2012). Where appropriate we shall use "13C-depleted/13C-enriched" as descriptors (but not for $\delta$13C values).

It would make reading easier if the authors replaced "d13C of deposit feeders respired CO2", which is grammatically awkward, with "d13C-CO2resp for deposition feeders". This abbreviation (or similar) could be used throughout the ms to improve readability.

Response: We will make the change throughout the ms.

Ln 14-16 and elsewhere: It is not immediately clear what is meant by "feeding regime"

Response: We shall explain it in parentheses.

Ln 26: "on field collection" – it is unclear what this means without having read the remainder of the paper

Response: We shall explain it in parentheses.

Ln 39: Remove 'past'

Response: Agreed.

Ln 47 onwards: Rework. The focus here on predators seems at odds with the focus of this manuscript on detritivores and deposit feeders.

Response: Agreed. We shall make the change to suit our focus.

Ln 48: Provide references for previous use of d13C-CO2 techniques

Response: We shall add references for the previous use of $\delta$13C-CO2 techniques, including Engel et al. (2009) and Carleton et al. (2004).

Ln 75-76: Unclear.

Response: We mean "few studies explore whether the increase in CO2 emission rates from sediment surface of mangroves is related to the feeding regime or the feeding habit of the benthos when they are included".

Ln 125-130: Some of this information would be better placed in the introduction

Response: Agreed. We shall include this information in the Introduction section.

Ln 126: Apparently crabs and gastropods were compared, but it seems likely there could be just as much different between the crab groups (ocypodids vs sesarmids) as between crabs and gastropods.

Response: There may be similar pattern on the differences. Ocypodids and sesarmids have been included as deposit feeders and detritivores in our analysis, respectively. We have compared the difference between crabs and gastropods (Line 221-3) as well as between deposit feeders and detritivores (Line 223-5), and thus have not directly compared the difference between ocypodids and sesarmids.

Ln 140: remove one mention of "collected"

Response: Agreed.

Ln 143: Unclear what is meant by "gas samples were collected separately".

Response: We mean gas samples were collected each time when the crabs were fed on different foods.

Ln 153-154: Presumably the samples were homogenised. Were the crabs dissected to remove tissues first?

Response: No, these groups of crabs were dried and then their weight measured.

The other group of crabs were dissected to remove tissues (Line 155-6) for isotopic analysis. Otherwise, if the tissues were removed first, the final weight of the crabs will be underestimated.

Ln 197&Ln 200: Specify what the groups were (e.g. crabs vs gastropods, or different crab groups vs gastropods?)

Response: Agreed. We shall show the groups as the reviewer indicated.

Ln 218-219 seems repeated in ln 219-220.

Response: We shall rearrange the sentences to avoid repeating the same information.

Ln 227-230: Somewhat misleading. The pattern is consistent, but the magnitude is different.

Response: We shall supplement the sentence to show there is a difference in magnitude.

Ln 245 (and elsewhere): reduce repetition, "CO2 production significantly increased with carapace length" (remove "there was a significant relationship. . ."_

Response: Agreed. We shall revise the sentences to avoid repetition in the context.

Ln 249-250: Sentence is unclear.

Response: We mean the application of the above relationship for intraspecific comparison may depend on the size measurement used. We shall correct the typo in the sentence.

Ln 259: It is not clear what is meant by 'categories'.

Response: We shall revise the latter part of the sentence to '. . .to infer different benthos foods being used'.

Section 4.1: The point of this section is unclear. It seems to mainly repeat the results, with no new inferences apparent. Can the authors specifically explain how d13C-

[Figure]

CO2resp can indicate dietary sources in some useful way?

Response: Agreed. We shall explain this point as shown in our response to the review's main concern 2.

Ln 294: The relevance of the similarity of d13C for fasting crabs and those fed on leaves is unclear

Response: We have shown that the similar $\delta$13C-CO2 for fasting crabs and those fed on leaves. Our result is supported by those of another study (Passey et al. 2005) (Line 296-7). It is likely some crabs (collected from the field) fed on both leaves and MPB, which have higher $\delta$13C (-24.8 $\pm$0.6 ‰ than leaves (-26.6 $\pm$0.3 ‰. After fasting, crabs metabolise stored C which have lower $\delta$13C than leaves. The higher $\delta$13C in their recent food and lower $\delta$13C in the stored C results in the $\delta$13C-CO2 for crabs similar to those fed on leaves.

Ln 297: Interesting that fractionation of d13C-CO2resp vs d13C-diet for crabs is similar to the range here, but gastropods seem to have far greater fractionation. Why? Where the diet of crabs was switched, is it possible that they were still using stored C (e.g. lipids), and this would have diluted the d13C value of the respired CO2 and affected results seen here?

Response: One possibility is that the different fractionation between crabs and gastropods arises from the differences in the amount of methane production by microorganisms in the digestive tract (Passey et al. 2005). Yes, the use of stored C when diet shifts would dilute the $\delta$13C-CO2.

Ln 327: Specify that C. perspicillata is a bat, and check the spelling of the species name.

Response: Agreed. We shall specify this point.

Ln 334: Check spelling of detritivore

Response: Agreed. We shall correct the typo.

Figure 1: There is one purple point in among the blue points – is this an error?

Response: Yes, we will remove it. Below is the modified figure.

Figure 4: The order of the bars (left to right) appears non-intuitive and does not match Figure 3. Also, the colours in this and other figures is unnecessary.

Response: We shall rearrange the bars to match Figure 3 and use the same colour for all the bars. Below is the modified figure.

Table 1: Is it possible to provide an error estimate for the fractionation values? E.g. d13C of individual animals – d13C of diet?

Response: Yes, we will add the standard error to the fractionation values. Below is the modified table.

References

Bond, A. L., & Hobson, K. A. (2012). Reporting stable-isotope ratios in ecology: recommended terminology, guidelines and best practices. Waterbirds, 35(2), 324-331.

Bui, T. H. H., & Lee, S. Y. (2014). Does 'You Are What You Eat' Apply to Mangrove Grapsid Crabs? PLoS One, 9(2), e89074. doi:10.1371/journal.pone.0089074

Engel, S., Lease, H. M., McDowell, N. G., Corbett, A. H., & Wolf, B. O. (2009). The use of tunable diode laser absorption spectroscopy for rapid measurements of the $\delta$13C of animal breath for physiological and ecological studies. Rapid Communications in Mass Spectrometry, 23(9), 1281-1286. doi:10.1002/rcm.4004

Carleton, S. A., Wolf, B. O., & Del Rio, C. M. (2004). Keeling plots for hummingbirds: a method to estimate carbon isotope ratios of respired CO2 in small vertebrates. Oecologia, 141(1), 1-6. Katz, L. C. (1980). Effects of burrowing by the fiddler crab, Uca pugnax (Smith). Estuarine and Coastal Marine Science, 11(2), 233-237.

Kristensen, E., Lee, S. Y., Mangion, P., Quintana, C. O., & Valdemarsen, T. (2017). Trophic discrimination of stable isotopes and potential food source partitioning by leaf‐eating crabs in mangrove environments. Limnology and Oceanography, 62(5), 2097-2112.

Ouyang, X., Lee, S. Y., & Connolly, R. M. (2017). Structural equation modelling reveals factors regulating surface sediment organic carbon content and CO2 efflux in a subtropical mangrove. Sci. Total. Environ., 578, 513-522. doi:10.1016/j.scitotenv.2016.10.218

Passey, B. H., Robinson, T. F., Ayliffe, L. K., Cerling, T. E., Sponheimer, M., Dearing, M. D., Roeder, B. L., & Ehleringer, J. R. (2005). Carbon isotope fractionation between diet, breath CO2, and bioapatite in different mammals. Journal of Archaeological Science, 32(10), 1459-1470.
* * *
[Figure]

**Fig. 1.** Figure 1

[Figure]

**Fig. 2.** Figure 4

Table 1

| Benthos taxa | | Diet | | Isotopic fractionation | |
|---|---|---|---|---|---|
| $\delta^{13}C\text{-}CO_2$ respiration for crabs | $\delta^{13}C\text{-}CO_2$ respiration for gastropods | $\delta^{13}C$ of MPB | $\delta^{13}C$ of yellow leaves | $\Delta^{13}C_{benthos\text{-}MPB}$ | $\Delta^{13}C_{benthos\text{-}leaves}$ |
| $-23.9\pm0.4‰$ | $-17.5\pm1.3‰$ | $-27.1\pm0.05‰$ | $-27.8\pm0.2‰$ | $3.2\pm0.04‰^a$, $9.6\pm0.35‰^b$ | $3.9\pm0.06‰^a$ |
| Feeding habit | | Diet | | Isotopic fractionation | |
| $\delta^{13}C\text{-}CO_2$ respiration deposit-feeders | $\delta^{13}C\text{-}CO_2$ respiration detritivores | $\delta^{13}C$ of MPB | $\delta^{13}C$ of yellow leaves | $\Delta^{13}C_{deposit\text{-}feeder\text{-}MPB}$ | $\Delta^{13}C_{detritivore\text{-}leaves}$ |
| $-19.8\pm0.8‰$ | $-24.7\pm0.3‰$ | $-27.1\pm0.05‰$ | $-27.8\pm0.2‰$ | $7.3\pm0.14‰$ | $3.1\pm0.07‰$ |

[a] crab, [b] gastropod

**Fig. 3.** Table 1